# Impact of Safe Water Programs on Water Treatment Practices of People Living with Human Immunodeficiency Virus, Ethiopia, 2008

**Sunkyung Kim [1,\*], Ciara E. O'Reilly [1], Sisay A. Abayneh [2], Achuyt Bhattarai [3], Jelaludin Ahmed [2], Alemayehu Mekonnen [2], Zainab Salah [1] and Rob Quick [1]**

1   Division of Foodborne, Waterborne, and Environmental Diseases, Centers for Disease Control and Prevention, Atlanta, GA 30333, USA; bwf1@cdc.gov (C.E.O.); pli1@cdc.gov (Z.S.); rxq1@cdc.gov (R.Q.)
2   Centers for Disease Control and Prevention, Addis Ababa 1230, Ethiopia; ipm3@cdc.gov (S.A.A.); hne9@cdc.gov (J.A.); hna8@cdc.gov (A.M.)
3   Enteric Diseases Epidemiology Branch, Centers for Disease Control and Prevention, Atlanta, GA 30333, USA; hij6@cdc.gov
\*   Correspondence: wox0@cdc.gov

**Abstract:** Household water chlorination has been shown to reduce diarrhea incidence among people living with Human Immunodeficiency Virus (PLHIV). Some HIV programs in Ethiopia previously provided a socially marketed chlorination product (brand name WuhaAgar) to prevent diarrhea. To evaluate the program, we compared WuhaAgar use and water treatment practices between 795 clients from 20 antiretroviral therapy (ART) clinics and 795 community members matched by age, sex, and neighborhood. Overall, 19% of study participants reported water treatment with WuhaAgar. Being an ART clinic client was associated with reported treatment of drinking water (matched odds ratios (mOR): 3.8, 95% confidence interval (CI): 2.9–5.0), reported current water treatment with WuhaAgar (mOR: 5.5, 95% CI 3.9–7.7), and bottles of WuhaAgar observed in the home (mOR: 8.8, 95% CI 5.4–14.3). Being an ART clinic client was also associated with reported diarrhea among respondents (mOR: 4.8, 95% CI 2.9–7.9) and household members (mOR: 2.8, 95% CI: 1.9–4.2) in the two weeks preceding the survey. Results suggest that promoting and distributing water chlorination products in ART clinics was effective in increasing access to and use of water treatment products among PLHIV. The positive association between ART clinic attendees and diarrhea likely resulted from the immunocompromised status of ART clinic clients.

**Keywords:** HIV; ART clinic; chlorination; diarrhea; PEPFAR

## 1. Introduction

In sub-Saharan Africa, diarrhea has been an important cause of morbidity and mortality in people living with Human Immunodeficiency Virus (PLHIV) [1–3]. Inexpensive interventions that prevent diarrhea, such as household water treatment and safe storage (HWTS), have been shown to protect PLHIV from some opportunistic infections (OIs) [4]. In recognition of findings from early studies, the President's Emergency Plan for AIDS Relief (PEPFAR) authorized the use of program funds for HWTS interventions for PLHIV [5,6].

Since 2003, the Centers for Disease Control and Prevention's (CDC) office in Ethiopia has partnered with other U.S. Government agencies under the direction of the U.S. Ambassador to implement interventions authorized by PEPFAR to strengthen Ethiopia's national response to the HIV/AIDS epidemic by providing financial and evidence-based technical assistance in the areas of HIV prevention, care, and treatment programs. In 2008, these interventions included a program to provide "preventive

care packages" (PCP) of proven interventions to 400,000 PLHIV to prevent OIs [7]. The first PCP program was implemented in Uganda in 2005 [8] and subsequently made available to populations of PLHIV in several other countries, including Ethiopia. The PCPs in Ethiopia included a dilute sodium hypochlorite solution (brand name WuhaAgar®, Addis Ababa, Ethiopia) designed for household water treatment and diarrhea prevention that was socially marketed in Ethiopia by Population Services International (PSI), an international nongovernmental organization (https://www.psi.org/).

Before the PCP program began, PSI distributed and socially marketed WuhaAgar in three regions of Ethiopia and implemented safe water programs with partners. The implementation approaches for WuhaAgar varied depending on the target group, but for HIV programs, WuhaAgar was distributed free, along with counseling on safe water and hygiene, and promotion of safe water storage. In preparation for the PCP Program, CDC conducted a baseline survey that included a component to assess the impact of WuhaAgar implementation activities in antiretroviral therapy (ART) programs (https://www.state.gov/pepfar/). The objectives of the survey were to determine the differences in household water treatment implementation among ART programs, describe hygiene behaviors and sanitation practices among PLHIV enrolled in ART clinics and among community members, and assess the effectiveness of ART clinics in the provision and promotion of water chlorination products to PLHIV.

## 2. Materials and Methods

### 2.1. Study Population

The evaluation was conducted from 8–18 December 2008, in three regions of Ethiopia (Amhara, Oromia, and the Southern Nations, Nationalities, and Peoples Region) where PSI socially marketed the WuhaAgar water disinfectant solution. As part of the HIV care and treatment program in Ethiopia, ART has been provided to PLHIV through public healthcare facilities since January 2005. In this evaluation, two groups of adults and children were included. One was the ART clinic client group (hereafter referred to as ART clinic clients) and the other a group of matched community members not enrolled in an ART clinic program, which served as a comparison group. Clients whose ART clinic appointment date fell within the dates of data collection were consecutively invited to participate in the evaluation. The comparison group included adults and children not participating in the HIV care and treatment program who were also living in communities within the catchment area of the healthcare facility of the selected clients.

### 2.2. Sample Size Calculation

To calculate the sample size required for the study, we assumed at least a 10% difference in WuhaAgar utilization rate between ART clinic clients (15%) and matched community members (5%), with a confidence level of 95%, power of 80%, design effect of four, and a non-response rate of 20%. Thus, 768 individuals were targeted for recruitment each among ART clinic clients and matched community members.

### 2.3. Sample Selection

Twenty healthcare facilities were randomly selected from a list of 75 healthcare facilities which provided ART services in the three regions. The number of ART clinic clients to be selected from each of the healthcare facilities was proportional to the client population of the facilities. In each selected healthcare facility, clients who were resident within the catchment area were consecutively enrolled until the pre-determined sample size was met. Once the main sections of the questionnaire were administered to enrolled clients at the ART clinic, a trained interviewer accompanied each enrolled ART clinic client to their household to complete the household observation section of the questionnaire. Community members were matched by age group (five-year interval), gender, and neighborhood to the enrolled clients. Thus, once the observations at the home of the ART clinic client were complete, the interviewer followed a randomization schedule to select the matched community member's household. The randomization

schedule required interviewers to first follow a pre-designated direction (left or right) while exiting from the ART clinic client's household and then, visit a pre-designated household which was determined by selecting a random number between 1 and 10. The schedule was randomly generated by investigators for each enrolled community member, and the pre-determined list was provided to all field interviewers. The interviewer visited successive households until the matching criteria to enroll community members were fulfilled.

### 2.4. Data Collection

The questionnaire included questions on demographic and socioeconomic characteristics; drinking water sources, storage, and treatment including WuhaAgar; and diarrhea incidence of responders and household members in the two weeks preceding the survey. Stored water was tested for free chlorine residual (FCR) in households where respondents reported using a chlorine product for water treatment. Water stored in households in which respondents reported not using chlorine was not tested in order to save time and resources. The evaluation questionnaires were translated and administered in Amharic by a trained interviewer to consenting study participants at their households. Parents or guardians responded on behalf of any participant who was younger than 18 years of age.

### 2.5. Statistical Analysis

Data were entered into Epi Info and all analyses were performed using SAS v9.4 (Cary, NC, USA). To examine any associations between ART clinic clients (=1 for client, =0 for community member) and selected characteristics, a univariable conditional logistic regression model was applied considering the matched case–control study design. We classified ART clinic clients into two groups: one group attending healthcare facilities that had stocks of WuhaAgar at the time of the survey (a proxy for an active water treatment program among clients) and a second group for the rest of the ART clinic clients. We then performed three additional subgroup analyses: first, we included ART clinic clients attending healthcare facilities with WuhaAgar stocks and matched community members; second, we included ART clinic clients attending healthcare facilities without WuhaAgar stocks and matched community members; and finally, we included ART clinic clients in healthcare facilities with WuhaAgar stocks vs. ART clinic clients in healthcare facilities without WuhaAgar. For the first two subgroups, the same analyses described above were conducted. For the last subgroup, complex survey analysis was conducted, where strata were the three geographic regions and primary sampling units were the healthcare facilities selected from each stratum. For FCR testing analysis, we classified households as non-chlorinators when respondents either reported no chlorine treatment or reported chlorinating stored water but had no detectable FCR in stored water; respondents who reported chlorinating water but had no water available for testing were treated as missing. Respondents with detectable FCR in stored water were classified as confirmed chlorinators.

### 2.6. Ethical Considerations

The study protocol was approved by the CDC Institutional Review Board (protocol number 5543) and the Ethiopian Public Health Association (protocol number 0030). Informed consent was obtained from all participants in Amharic and for participants less than 18 years old, proxy consent was obtained from the parent or guardian.

## 3. Results

### 3.1. Demographics and Household Characteristics

Of the 1590 total study participants in the two groups, the mean age was 33.9 years (Interquartile range: 26–40 years) and 73% were female. Twenty-eight percent of all participants reported not having any formal education, and 60% rented their dwelling (Table 1).

**Table 1.** Demographic and household characteristics of respondents, N [1] (%), of antiretroviral therapy (ART) clients (*n* = 795) and community members (*n* = 795), Ethiopia, 2008.

| Characteristic | All N (%) | ART Clients N (%) | Community Members N (%) |
|---|---|---|---|
| Earned household income source | | | |
| Yes | 728 (47) | 289 (37) | 439 (57) |
| No | 833 (53) | 497 (63) | 336 (43) |
| Education | | | |
| No formal schooling | 442 (28) | 206 (26) | 236 (30) |
| ≤Primary | 615 (39) | 259 (33) | 356 (45) |
| >Primary | 533 (33) | 330 (41) | 203 (25) |
| People living in household | | | |
| ≤ 4 | 948 (60) | 528 (66) | 420 (53) |
| > 4 | 641 (40) | 267 (34) | 374 (47) |
| Rent or own a home | | | |
| Rent | 956 (60) | 538 (68) | 418 (53) |
| Own | 585 (37) | 216 (27) | 369 (46) |
| Other | 47 (3) | 40 (5) | 7 (1) |
| Household owns | | | |
| Electricity | 1502 (94) | 740 (93) | 762 (96) |
| Radio | 1229 (77) | 565 (71) | 664 (83) |
| Television | 728 (46) | 288 (36) | 440 (55) |
| Mobile phone | 480 (30) | 164 (21) | 316 (40) |
| Refrigerator | 154 (10) | 41 (5) | 113 (14) |
| Bicycle | 76 (5) | 25 (3) | 51 (6) |
| Horse/mule | 23 (1) | 17 (2) | 6 (1) |

[1] N do not add up to total due to missing values for some variables.

Compared to community members, a higher percentage of ART clinic clients reported no earned household income (43% vs. 63%, respectively) and rented (rather than owned) their home (53% vs. 68%); a lower percentage of ART clinic clients reported living in a house shared with >4 people and (with the exception of horse/mule) possessing each item on a list of seven household assets.

*3.2. Drinking Water Source, Storage, and Treatment*

Almost all participants (99%) had access to an improved water supply, 86% thought their water was safe to drink, 89% were observed to use improved drinking water storage containers, and 29% reported they treated their water (Table 2). A higher percentage of ART clinic clients than matched community controls reported treating their water with chlorine products (30% vs. 8%). Believing drinking water was unsafe (matched odds ratio (mOR) = 1.6, 95% confidence interval (CI) = 1.1–2.1), storing water in improved containers (mOR = 2.7, 95% CI = 1.7–4.3), reported drinking water treatment (mOR = 3.8, 95% CI = 2.9–5.0), and reported use of chlorine products to treat water (mOR = 5.6, 95% CI = 4.0–7.9) were all positively associated with the ART clinic client group. Compared to non-chlorinators (i.e., persons who either reported non-chlorination or reported chlorination but had stored water with a negative FCR), confirmed chlorinators had greater odds of being in the ART clinic client group (mOR = 13.3; 95% CI = 4.8–36.6).

**Table 2.** Household drinking water source, storage, treatment, and diarrhea incidence, N [1] (%), of ART clients (*n* = 795) and community members (*n* = 795), and matched odds ratio (mOR) of being ART client, Ethiopia, 2008.

| Characteristic | All N (%) | ART Clients N (%) | Community Members N (%) | mOR (95% CI) |
|---|---|---|---|---|
| Drinking water source [2] | | | | |
| Improved | 1568 (99) | 784 (99) | 784 (99) | Ref [3] |
| Unimproved | 22 (1) | 11 (1) | 11 (1) | 1.0 (0.4–2.9) |
| Think drinking water safe | | | | |
| Yes | 1330 (86) | 652 (84) | 678 (88) | Ref [3] |
| No | 216 (14) | 127 (16) | 89 (12) | 1.6 (1.1–2.1) |
| Observed storage of drinking water [4] | | | | |
| Improved | 1410 (89) | 722 (91) | 688 (87) | 2.7 (1.7–4.3) |
| Unimproved | 180 (11) | 73 (9) | 107 (13) | Ref [3] |
| Treat drinking water | | | | |
| Yes | 451 (29) | 320 (41) | 131 (17) | 3.8 (2.9–5.0) |
| No | 1130 (71) | 469 (59) | 661 (83) | Ref [3] |
| Usual practice for drinking water treatment | | | | |
| Use chlorine products [5] | 301 (19) | 238 (30) | 63 (8) | 5.6 (4.0–7.9) |
| Others | 1289 (81) | 557 (70) | 732 (92) | Ref [3] |
| Treated current drinking water with WuhaAgar | | | | |
| Yes | 303 (19) | 242 (30) | 61 (8) | 5.5 (3.9–7.7) |
| No | 1287 (81) | 553 (70) | 734 (92) | Ref [3] |
| Observed bottle of WuhaAgar at home | | | | |
| Yes | 193 (12) | 166 (21) | 27 (3) | 8.8 (5.4–14.3) |
| No | 1396 (88) | 628 (79) | 768 (97) | Ref [3] |
| Detectable FCR in stored water | | | | |
| Yes | 61 (4) | 55 (7) | 6 (1) | 13.3 (4.8–36.6) |
| No [6] | 1485 (96) | 703 (93) | 782 (99) | Ref [3] |
| Ever received WuhaAgar for free | | | | |
| Yes | 232 (15) | 213 (27) | 19 (2) | 20.0 (10.6–37.7) |
| No | 1352 (85) | 582 (73) | 770 (98) | Ref [3] |
| Ever received counseling on how to use WuhaAgar | | | | |
| Yes | 471 (30) | 309 (39) | 162 (20) | 2.6 (2.0–3.3) |
| No | 1117 (70) | 486 (61) | 631 (80) | Ref [3] |
| Have had diarrhea <2 weeks | | | | |
| Yes | 118 (7) | 95 (12) | 23 (3) | 4.8 (2.9–7.9) |
| No | 1472 (93) | 700 (88) | 772 (97) | Ref [3] |

**Table 2.** *Cont.*

| Characteristic | | All N (%) | ART Clients N (%) | Community Members N (%) | mOR (95% CI) |
|---|---|---|---|---|---|
| Household member had diarrhea <2 weeks | | | | | |
| | Yes | 136 (9) | 98 (12) | 38 (5) | 2.8 (1.9–4.2) |
| | No | 1454 (91) | 697 (88) | 757 (95) | Ref [3] |

[1] N do not add up to total due to missing for some variables. [2] Improved includes piped water, protected dug well/spring, tube well, borehole, rainwater collection/burka; Unimproved includes bottled water, unprotected dug well, unprotected spring, tanker-truck, cart with small tank, surface water. [3] Referent. [4] Improved includes container with cover/closed bucket, covered bucket with tap, narrow mouth container/jerrycan; Unimproved includes open container/bucket, traditional clay pot. [5] WuhaAgar, PuR, Chlorine tablet/bleach. [6] Respondents who either reported non-use of chlorine products for water treatment or reported chlorine use but had no detectable FCR in stored water.

### 3.3. Coverage of WuhaAgar

Most participants (80%) had heard of WuhaAgar. Overall, less than one-fifth of participants reported that their current drinking water was treated with WuhaAgar (19%) or had a WuhaAgar bottle observed at home (12%) (Table 2). There was a positive association between the ART clinic client group and reported current drinking water treatment with WuhaAgar (mOR = 5.5, 95% CI = 3.9–7.7), having a bottle of WuhaAgar observed in their homes (mOR = 8.8, 95% CI = 5.4–14.3), reporting having ever received WuhaAgar for free (mOR = 20.0 95% CI= 10.6–37.7), and receiving counseling on the product (mOR = 2.6, 95% CI = 2.0–3.3). In a subgroup analysis, ART clinic clients attending healthcare facilities that had stocks of WuhaAgar (N = 211, four healthcare facilities) had higher rates of improved water storage (96% vs. 87%), reported drinking water treatment (51% vs. 37%), and having a bottle of WuhaAgar bottle observed in the home (42% vs. 13%) than ART clinic clients attending healthcare facilities without WuhaAgar stocks (Table 3). Using improved water storage containers (OR = 3.7, 95% CI = 1.6–8.4), reporting water treatment (mOR = 1.0, 95% CI = 1.0–1.1), and having bottles of WuhaAgar observed in the home (mOR = 4.6, 95% CI = 1.6–13.4) were positively associated with ART clinic clients attending healthcare facilities with WuhaAgar stock.

**Table 3.** Household drinking water source, storage, treatment, and diarrhea incidence, N [1] (%), of ART clients attending clinics with stock of WuhaAgar (*n* = 211) and other ART clients (*n* = 584), and odds ratio (OR) of being ART client in clinics with stock of WuhaAgar, Ethiopia, 2008.

| Characteristic | | ART Clients with Stock of WuhaAgar N (%) | Other ART Clients N (%) | OR (95% CI) |
|---|---|---|---|---|
| Drinking water source [2] | | | | |
| | Improved | 210 (99) | 574 (98) | Ref [3] |
| | Unimproved | 1 (1) | 10 (2) | 0.3 (0.1–1.0) |
| Think drinking water safe | | | | |
| | Yes | 168 (80) | 484 (85) | Ref [3] |
| | No | 41 (20) | 86 (15) | 1.4 (0.7–2.8) |
| Observed storage of drinking water [4] | | | | |
| | Improved | 177 (96) | 450 (87) | 3.7 (1.6–8.4) |
| | Unimproved | 7 (4) | 66 (13) | Ref [3] |
| Treat drinking water | | | | |
| | Yes | 106 (51) | 214 (37) | 1.0 (1.0–1.1) |
| | No | 102 (49) | 367 (63) | Ref [3] |
| Usual practice for drinking water treatment | | | | |
| | Use chlorine products [5] | 92 (44) | 146 (25) | 2.3 (0.7–8.2) |
| | Others | 119 (56) | 438 (75) | Ref [3] |

**Table 3.** *Cont.*

| Characteristic | | ART Clients with Stock of WuhaAgar N (%) | Other ART Clients N (%) | OR (95% CI) |
|---|---|---|---|---|
| Treated current drinking water with WuhaAgar | | | | |
| | Yes | 97 (46) | 145 (25) | 2.6 (0.7–10.1) |
| | No | 114 (54) | 439 (75) | Ref [3] |
| Observed bottle of WuhaAgar at home | | | | |
| | Yes | 88 (42) | 78 (13) | 4.6 (1.6–13.4) |
| | No | 123 (58) | 505 (87) | Ref [3] |
| Ever received WuhaAgar for free | | | | |
| | Yes | 101 (48) | 112 (19) | 3.9 (0.7–22.2) |
| | No | 110 (52) | 472 (81) | Ref [3] |
| Ever received counseling on how to use WuhaAgar | | | | |
| | Yes | 107 (51) | 202 (35) | 1.9 (0.6–6.7) |
| | No | 104 (49) | 382 (65) | Ref [3] |
| Have had diarrhea <2 weeks | | | | |
| | Yes | 25 (12) | 70 (12) | 0.9 (0.7–1.5) |
| | No | 186 (88) | 514 (88) | Ref [3] |
| Household member had diarrhea <2 weeks | | | | |
| | Yes | 29 (14) | 69 (12) | 1.2 (0.8–1.7) |
| | No | 182 (86) | 515 (88) | Ref [3] |

[1] N do not add up to total due to missing for some variables. [2] Improved includes piped water, protected dug well/spring, tube well, borehole, rainwater collection/burka; Unimproved includes bottled water, unprotected dug well, unprotected spring, tanker-truck, cart with small tank, surface water. [3] Referent. [4] Improved includes container with cover/closed bucket, covered bucket with tap, narrow mouth container/jerrycan; Unimproved includes open container/bucket, traditional clay pot. [5] WuhaAgar, PuR, Chlorine tablet/bleach.

*3.4. Incidence of Diarrhea*

Less than one-tenth of all participants reported they had diarrhea (7%) or had any household member with diarrhea (9%) in the previous two weeks (Table 2). There was a positive association between ART clinic clients and reporting an episode of diarrhea in the previous two weeks (mOR = 4.8, 95% CI = 2.9–7.9) and reporting diarrhea among household members in the previous two weeks (mOR = 2.8, 95% CI = 1.9–4.2). Similar associations were observed in two subgroup analyses that compared reports of diarrhea between ART clients in healthcare facilities with WuhaAgar stocks and matched community members, and between ART clients in healthcare facilities without WuhaAgar stocks and matched community members (Table 4).

Overall, these associations were stronger among ART clinic clients in healthcare facilities with WuhaAgar stock than in ART clinic clients in healthcare facilities without WuhaAgar. When these models were adjusted for reported current water treatment, the strength of the associations described above were unchanged.

**Table 4.** Drinking water source, storage, treatment, and diarrhea incidence, N [1] (%), of ART clients attending clinics with stock of WuhaAgar (*n* = 211) and their matched community members (*n* = 211), other ART clients (*n* = 584) and their matched community members (*n* = 584), and matched odds ratio (mOR) of being the corresponding ART client, Ethiopia, 2008.

| Characteristic | | ART Clients with Stock of WuhaAgar N (%) | Community Members N (%) | mOR (95% CI) | Other ART Clients N (%) | Community Members N (%) | mOR (95% CI) |
|---|---|---|---|---|---|---|---|
| Drinking water source [2] | | | | | | | |
| | Improved | 210 (99) | 211 (100) | Ref | 574 (98) | 573 (98) | Ref [3] |
| | Unimproved | 1 (1) | 0 (0) | 1.0 (0.1-Inf) | 10 (2) | 11 (2) | 0.9 (0.3–2.6) |
| Think drinking water safe | | | | | | | |
| | Yes | 168 (80) | 190 (95) | Ref | 484 (85) | 488 (86) | Ref [3] |
| | No | 41 (20) | 10 (5) | 4.2 (2.0–8.7) | 86 (15) | 79 (14) | 1.1 (0.8–1.6) |
| Observed storage of drinking water [4] | | | | | | | |
| | Improved | 177 (96) | 159 (94) | 2.7 (0.7–10.1) | 450 (87) | 402 (81) | 2.7 (1.6–4.5) |
| | Unimproved | 7 (4) | 11 (6) | Ref | 66 (13) | 96 (19) | Ref [3] |
| Treat drinking water | | | | | | | |
| | Yes | 106 (51) | 27 (13) | 9.0 (4.7–17.3) | 214 (37) | 104 (18) | 2.9 (2.2–4.0) |
| | No | 102 (49) | 184 (87) | Ref | 367 (63) | 477 (82) | Ref [3] |
| Usual practice for drinking water treatment | | | | | | | |
| | Use chlorine products [5] | 92 (44) | 13 (6) | 16.8 (6.9–41.4) | 146 (25) | 50 (9) | 3.9 (2.6–5.7) |
| | Others | 119 (56) | 198 (94) | Ref | 438 (75) | 534 (91) | Ref [3] |
| Treated current drinking water with WuhaAgar | | | | | | | |
| | Yes | 97 (46) | 11 (5) | 18.2 (7.4–44.8) | 145 (25) | 50 (9) | 3.7 (2.6–5.4) |
| | No | 114 (54) | 200 (95) | Ref | 439 (75) | 534 (91) | Ref [3] |
| Observed bottle of WuhaAgar at home | | | | | | | |
| | Yes | 88 (42) | 5 (2) | 42.5 (10.7–172.7) | 78 (13) | 22 (3.8) | 4.5 (2.6–7.8) |
| | No | 123 (58) | 206 (98) | Ref | 505 (87) | 562 (96) | Ref [3] |
| Ever received WuhaAgar for free [6] | | | | | | | |
| | Yes | 101 (48) | 5 (2) | 48.5 (11.9–196.7) | 112 (19) | 14 (2) | 12.9 (6.3–26.4) |
| | No | 110 (52) | 204 (98) | Ref | 472 (81) | 566 (98) | Ref [3] |

**Table 4.** *Cont.*

| Characteristic | | ART Clients with Stock of WuhaAgar N (%) | Community Members N (%) | mOR (95% CI) | Other ART Clients N (%) | Community Members N (%) | mOR (95% CI) |
|---|---|---|---|---|---|---|---|
| Ever received counseling on how to use WuhaAgar | | | | | | | |
| | Yes | 107 (51) | 38 (18) | 5.3 (3.1–9.1) | 202 (35) | 124 (21) | 2.0 (1.5–2.6) |
| | No | 104 (49) | 173 (82) | Ref | 382 (65) | 458 (79) | Ref [3] |
| Have had diarrhea <2 weeks | | | | | | | |
| | Yes | 25 (12) | 3 (1) | 8.3 (2.5–27.6) | 70 (12) | 20 (3) | 4.1 (2.4–7.1) |
| | No | 186 (88) | 208 (99) | Ref | 514 (88) | 564 (97) | Ref [3] |
| Household member had diarrhea <2 weeks | | | | | | | |
| | Yes | 29 (14) | 6 (3) | 6.8 (2.4–19.3) | 69 (12) | 32 (5) | 2.3 (1.5–3.5) |
| | No | 182 (86) | 205 (97) | Ref | 515 (88) | 552 (95) | Ref [3] |

[1] N do not add up to total due to missing for some variables. [2] Improved includes piped water, protected dug well/spring, tube well, borehole, rainwater collection/burka; Unimproved includes bottled water, unprotected dug well, unprotected spring, tanker-truck, cart with small tank, surface water. [3] Referent. [4] Improved includes container with cover/closed bucket, covered bucket with tap, narrow mouth container/jerrycan; Unimproved includes open container/bucket, traditional clay pot. [5] WuhaAgar, PuR, Chlorine tablet/bleach. [6] From hospital, clinic, non-governmental organization, or else.

## 4. Discussion

The findings from this study suggest that ART clinic clients were more likely than matched community members to report chlorination of household water, to have detectable FCR in stored water, to have a WuhaAgar bottle at home, and to store water in improved containers. Of note, clients who attended ART clinics in healthcare facilities that stocked WuhaAgar were more likely to report that they engaged in water treatment practices at home than ART clinic clients in healthcare facilities that did not stock WuhaAgar. These findings were consistent with other studies that have suggested that healthcare facilities are an effective venue for promoting household water treatment [9–11].

Despite reported higher levels of WuhaAgar use and confirmed higher rates of chlorination at home among ART clinic clients than matched community members, the reported WuhaAgar use rate for ART clinic clients was modest at 30%, and was even lower, at 7%, for confirmed use of hypochlorite. Previous studies have found that confirmed chlorine use tends to be lower than reported use [12–14]. Among possible explanations for modest water treatment rates were findings that one-third of ART clinic clients had no formal education, and over 60% had no regular income. Previous studies have shown that users who purchased point-of-use household water treatment products were typically more educated, from urban areas, and employed [8,15], but also that healthcare facility-based distribution can help reduce the barriers posed by lower education levels, rural residence, and underemployment [10]. In addition, free distribution from healthcare facilities may have helped overcome economic barriers faced by PLHIV, whose illness may inhibit their ability to work.

Although we did not specifically address stigma as a potential barrier to use of WuhaAgar provided by ART clinics (that is, that the presence of WuhaAgar in the home might have the undesired impact of identifying project participants as HIV-infected), higher use of the product among ART clinic clients than matched community members suggests that stigma was not a problem for the participants of water treatment programs. This finding can at least partly be explained by high product awareness in Ethiopia attributable to previous WuhaAgar social marketing programs in the general population.

An unexpected finding in this survey was that more diarrhea episodes were reported in ART clinic clients and their household members than matched community controls, despite differential use of WuhaAgar. Results of prior research on other populations in Africa suggest that diarrhea incidence can be reduced among PLHIV through the use of household water chlorination products, especially among persons receiving cotrimoxazole prophylaxis [3]. One possible explanation for the discrepant results in this study was the modest reported use of water treatment products compared to higher confirmed use rates in other studies. When statistical models in this study adjusted for current water treatment, there were no appreciable differences in the strength of association, suggesting that diarrhea was not associated with water treatment and that other risk factors were more important. In the absence of an effect of water treatment on diarrhea, another potential explanation for the contradictory findings is that higher diarrhea rates would be expected among populations with immune deficiencies than in immunocompetent community members [3]. ART clinic clients would have also likely made more regular clinic visits than community members to obtain supplies of antiretroviral medicines, which would have provided more opportunities to report diarrhea. Previous studies have documented low use of healthcare facilities in the general population in Ethiopia [16–18].

Although this study was intended to be a baseline for a controlled trial examining the health impact of a basic PCP containing health products funded by PEPFAR among ART clients in 20 healthcare facilities, a shortage in funding limited the subsequent health impact evaluation to two hospitals [19]. The simple and relatively inexpensive water quality interventions in the PCP (WuhaAgar, a flocculant/disinfectant product, and safe water storage containers) were confirmed in intervention ART clients to be used at a substantially higher rate, and to result in a lower likelihood of reported illness and healthcare facility visits for illness, and better overall health scores, than in control ART clients [19]. Similar interventions were also reported to decrease diarrhea risk in HIV-infected children in Kenya [20] and adults in Uganda [3].

Although this evaluation was conducted over a decade ago, the results remain relevant. Findings from more recent studies in several countries, including Ethiopia, serve as reminders that progress toward improving the health of PLHIV in low resource settings can be enhanced by interventions that prevent and reduce the risk of OIs [4,20]. Despite the ongoing risk of OIs, reductions in PEPFAR funding have greatly reduced the scope of potential health interventions for PLHIV in developing countries [21,22]. For example, in Ethiopia, the PCP program ended in 2012 because of PEPFAR funding cuts. Although access to safely managed or basic water supplies in Ethiopia increased from an estimated 30% of households in 2008 to 41% in 2017, coverage remains poor [23]. The lack of safe drinking water remains a public health threat for developing country populations, including PLHIV [24,25]. In Ethiopia, at least three water treatment products are commercially available, which may be a response to consumer perceptions of a need for methods to make drinking water safe [26,27].

This study had several important limitations. First, because of time and resource constraints, our sample only included ART clinics in 16 hospitals and 4 health centers. Smaller healthcare facilities were, therefore, underrepresented. Second, for logistical reasons, we enrolled participants who lived within a radius of 10 km from each healthcare facility and were unable to discern water, sanitation, and hygiene characteristics of more remote populations. Third, our inclusion of respondents who reported no chlorine treatment with confirmed non-chlorinators potentially underestimated the percentage of the population that used chlorine, which could have biased our results. However, over-reporting of water treatment is typically more common than under-reporting (perhaps as a result of social desirability bias) [12–14]. Fourth, this study may have been underpowered because of the reduced sample size in subgroup analyses comparing clients attending ART clinics with stocks of WuhaAgar with clients attending other ART clinics. Finally, the cross-sectional evaluation design was not as strong methodologically as a longitudinal study would have been, particularly to infer possible causality between clinic client status and diarrhea incidence.

## 5. Conclusions

The results of this survey suggest that safe water programs in healthcare facilities in Ethiopia resulted in greater utilization of point-of-use water treatment products among clients of ART programs than among community members living in the same healthcare facility catchment areas. These findings are consistent with the results of other healthcare facility-based safe water programs among HIV-infected [8] and non-infected populations [9,10], and support the use of healthcare facilities for distribution of evidence-based health products in programs that serve PLHIV. The lack of impact of these interventions on diarrheal disease in this survey was not consistent with previous work examining similar interventions and stresses the importance of maximizing adherence to water treatment interventions among at-risk populations to achieve the desired impact.

**Author Contributions:** R.Q., C.E.O., S.A.A. and A.B. designed the study; S.K. and Z.S. analyzed the data; C.E.O., S.A.A., A.B., J.A., A.M. and R.Q. contributed to data collection and interpretation; all authors contributed to the writing and editing of the manuscript. All authors have read and agreed to the published version of the manuscript.

**Funding:** Funding for this evaluation was provided by the US President's Emergency Plan for AIDS Relief, Global AIDS Program, Centers for Disease Control and Prevention, Ethiopia; and the U.S. Agency for International Development.

**Acknowledgments:** The investigators thank the following groups for their contribution to the evaluation: Population Services International; Addis Continental Institute of Public Health; Ethiopian Ministry of Health, Disease Prevention and Control Office; Ethiopian Ministry of Health, Hygiene and Environmental Health Department; The Ministry of Water Resources of Ethiopia, Water Supply and Sanitation Projects; Federal HIV/AIDS Prevention and Control Office Ethiopia, Palliative Care Task Force; Abt Associates; PLHIV associations; and the Ethiopian Public Health Association, and Robert M. Hoekstra, Kathleen Wannemuehler, and Tom Taylor, Division, Foodborne, Bacterial And Mycotic Diseases, Centers for Disease Control and Prevention, Atlanta, GA, USA. We thank the study participants, and the management, medical staff, and PEPFAR implementing partners at Adama Hospital, Nekemet Hospital, Ambo Hospital, Bishoftu Hospital, Fiche Hospital, Mojo Health Center, Woliso Hospital, Goba Hospital, Dilla Hospital, Attat Hospital, Arbaminch Hospital, Dessie Hospital, Finoteselam Hospital, Gonder Hospital, Woldia Hospital, Bure Health Center, Mersa Health Center, Kombolcha Health Center, Debre Markos Hospital, and Debre Brehan Hospital.

**Conflicts of Interest:** The authors declare no conflict of interest.

**Disclaimer:** The contents of this paper are solely the responsibility of the authors and do not necessarily reflect the official views of the Centers for Disease Control and Prevention (CDC).

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
