# Peer review of "Impact of Safe Water Programs on Water Treatment Practices of People Living with Human Immunodeficiency Virus, Ethiopia, 2008"

_water, doi:10.3390/w12113261_

Round 1
Reviewer 1 Report
The work is excellent, and merits publication. However, there are two methodological points which, while acknowledged currently, should be expanded on to render the findings more useful.
- The finding that PLHIV undergoing treatment have a higher incidence rate of diarrhea, despite higher use of in-home water chlorination, is not surprising, given their immunocompromised status. Are there any in-country measures of baseline incidence of diarrhea among PLHIV or all citizens of Ethiopia?
- The data are 10+ years old. Some context as to the evolution of these programs and the state of water and sanitation in country would be useful. Have there been overall gains in water and sanitation coverage in Ethiopia? Changes in PLHIV incidence? Are these programs still in operation?
Author Response
1. The finding that PLHIV undergoing treatment have a higher incidence rate of diarrhea, despite higher use of in-home water chlorination, is not surprising, given their immunocompromised status. Are there any in-country measures of baseline incidence of diarrhea among PLHIV or all citizens of Ethiopia?
: We were unable to obtain data on the incidence of diarrheal illness among PLHIV in Ethiopia. Among children <5 years old, the 2-week prevalence of diarrhea was estimated to be 13% in 2011 and 12% in 2016. Because diarrhea rates in children have not changed in the recent past and we lack baseline diarrhea data in adults or PLHIV, we have left out these data.
2. The data are 10+ years old. Some context as to the evolution of these programs and the state of water and sanitation in country would be useful. Have there been overall gains in water and sanitation coverage in Ethiopia? Changes in PLHIV incidence? Are these programs still in operation?
We have added program context to the discussion section (lines 267-268): “In Ethiopia, the PCP program ended in 2012 because of PEPFAR funding cuts.” We have added water supply context to the discussion section (lines 268-270): “Although access to safely managed or basic water supplies in Ethiopia increased from an estimated 30% of households in 2008 to 41% 2017, coverage remains poor.” Finally, in the same paragraph (lines 271-273), we noted current state of water treatment availability: “In Ethiopia, at least three water treatment products are commercially available, which may be a response to consumer perceptions of a need for methods to make drinking water safe.”
Reviewer 2 Report
This paper deals with an overview concerning marketed chlorination product to prevent diarrhea, reported in a HIV programs previously provided in Ethiopia.
The authors try to summarize an interesting and relevant topic related to water safety. The manuscript appears interesting and provides useful information concerning the topic. I think that the readability of the manuscript could be improved in the interest of consistency by better introducing the specific field (more specific references). The title should not contain acronyms and should be more incisive. Acronyms should be better specified and used in the whole text, Table 3 and 4 should be better organized. Remarkably, the authors should better distinguish in the whole manuscript new data from old data by clarifying how this report could help to shed light in this scientific field. As a whole, in my opinion, after revision this article could be considered for publication.
Author Response
- This paper deals with an overview concerning marketed chlorination product to prevent diarrhea, reported in a HIV programs previously provided in Ethiopia. The authors try to summarize an interesting and relevant topic related to water safety. The manuscript appears interesting and provides useful information concerning the topic. I think that the readability of the manuscript could be improved in the interest of consistency by better introducing the specific field (more specific references).
: We added a line in the introduction (lines 43-45) that provided context for the preventive care package program: “The first PCP program was implemented in Uganda in 2005 [8] and subsequently made available to populations of PLHIV in several other countries, including Ethiopia.”
- The title should not contain acronyms and should be more incisive. Acronyms should be better specified and used in the whole text.
:As requested, we have changed the title of the paper to “Impact of safe water programs on water treatment practices of people living with Human Immunodeficiency Virus, Ethiopia, 2008.” We also have gone through the paper and made sure that the first use of each acronym was spelled out.
- Table 3 and 4 should be better organized.
:To clarify all 4 tables, we have included N (%) at the top of each column with relevant data. Table 3 and 4 are organized in standard fashion, but to better organize the appearance of the tables we have expanded the width of the columns. Table 4 was changed from portrait to landscape orientation to better fit the changed formatting.
- Remarkably, the authors should better distinguish in the whole manuscript new data from old data by clarifying how this report could help to shed light in this scientific field.
:We have revised the following paragraph in the discussion (lines 264-275) to demonstrate the relevance of 2008 evaluation data to the current situation:
“Although this evaluation was conducted over a decade ago, the results remain relevant. Findings from more recent studies in several countries, including Ethiopia, serve as reminders that progress toward improving the health of PLHIV in low resource settings can be enhanced by interventions that prevent and reduce the risk of OIs [4,20]. Despite the ongoing risk of OIs, reductions in PEPFAR funding have greatly reduced the scope of potential health interventions for PLHIV in developing countries [21,22]. For example, in Ethiopia, the PCP program ended in 2012 because of PEPFAR funding cuts. Although access to safely managed or basic water supplies in Ethiopia increased from an estimated 30% of households in 2008 to 41% in 2017 [25], coverage remains poor. The lack of safe drinking water remains a public health threat for developing country populations, including PLHIV [23, 24]. In Ethiopia, at least three water treatment products are commercially available, which may be a response to consumer perceptions of a need for methods to make drinking water safe [26,27].”